# Large Pericardial Effusion—Diagnostic and Therapeutic Options, with a Special Attention to the Role of Prolonged Pericardial Fluid Drainage

**DOI:** 10.3390/diagnostics12061453

**Published:** 2022-06-13

**Authors:** Małgorzata Dybowska, Monika Szturmowicz, Katarzyna Błasińska, Juliusz Gątarek, Ewa Augustynowicz-Kopeć, Renata Langfort, Paweł Kuca, Witold Tomkowski

**Affiliations:** 1I Department of Lung Diseases, National Tuberculosis and Lung Diseases Research Institute, 01-138 Warsaw, Poland; monika.szturmowicz@gmail.com (M.S.); w.tomkowski@igichp.edu.pl (W.T.); 2Department of Radiology, National Tuberculosis and Lung Diseases Research Institute, 01-138 Warsaw, Poland; kasiabp67@gmail.com; 3Department of Thoracic Surgery, National Tuberculosis and Lung Diseases Research Institute, 01-138 Warsaw, Poland; j.gatarek@igichp.edu.pl; 4Department of Microbiology, National Tuberculosis and Lung Diseases Research Institute, 01-138 Warsaw, Poland; e.kopec@igichp.edu.pl; 5Department of Pathology, National Tuberculosis and Lung Diseases Research Institute, 01-138 Warsaw, Poland; renata.langfort@gmail.com; 6Department of Internal Medicine, Endocrinology and Diabetology, Medical University of Warsaw, 61 Zwirki i Wigury Street, 02-091 Warsaw, Poland; pawel.jan.kuca@gmail.com

**Keywords:** large pericardial effusion, cardiac tamponade, pericardiocentesis, pericardioscopy, neoplastic pericardial effusion, purulent pericardial effusion, tuberculous pericardial effusion

## Abstract

Background: Large pericardial effusion (LPE) is associated with high mortality. In patients with cardiac tamponade or with suspected bacterial etiology of pericardial effusion, urgent pericardial decompression is necessary. Aim: The aim of the present retrospective study was to assess the short-term results of pericardial decompression combined with prolonged drainage in LPE. Material: This study included consecutive patients with LPE who had been treated with pericardial fluid drainage between 2007 and 2017 in the National Tuberculosis and Lung Diseases Research Institute. Methods: Echocardiographic examination was used to confirm LPE and the signs of cardiac tamponade. Pericardiocentesis or surgical decompression were combined with pericardial fluid (PF) drainage. Short-term effectiveness of therapy was defined as less than 5 mm of fluid behind the left ventricular posterior wall in echocardiography. Results: The analysis included 74 patients treated with pericardial fluid drainage (33 female and 41 male), mean age 58 years, who underwent pericardial decompression. Out of 74 patients, 26 presented with cardiac tamponade symptoms. Pericardiocentesis was performed in 18 patients and pericardiotomy in 56 patients. Median PF drainage duration was 13 days. In 17 out of 25 patients with neoplastic PF, intrapericardial cisplatin therapy was implemented. In 4 out of 49 patients with non-malignant PF, purulent pericarditis was recognized and intrapericardial fibrinolysis was used. Short-term effectiveness of the therapy was obtained in all of patients. Non-infective complications were noted in 16% of patients and infective ones in 10%. Conclusion: Pericardial decompression combined with prolonged PF drainage was safe and efficient method of LPE treatment.

## 1. Introduction

Large pericardial effusion (LPE) is defined as an echo-free space exceeding 20 mm, measured behind the posterior wall of the left ventricle on echocardiographic examination [1,2,3,4].

The most frequent causes of LPE are neoplastic disease and viral pericarditis [1,3,5]. An increasing role of iatrogenic causes of LPE has been observed recently [1,3].

In developing countries, where tuberculosis is endemic, it is the dominant cause of pericarditis and includes 60% of LPE cases [6,7].

New diagnostic strategies concerning the management of LPE have been proposed in the ESC guidelines in 2015 for the selection of high-risk patients who are candidates for urgent pericardial fluid drainage [1,8].

Pericardial drainage is recommended in patients with cardiac tamponade, with suspected bacterial or neoplastic etiology of pericardial effusion, as well as in those who present with chronic (>3 months) LPE [1].

Pericardiocentesis and pericardiotomy are the methods of choice [1,2,9].

Extended pericardial drainage prevents the recurrence of PF and enables intrapericardial therapy if needed; nevertheless, it is not recommended as the standard procedure in recent ESC guidelines. Moreover, the optimal duration of pericardial drainage and its efficacy and safety are unknown.

## 2. Aim

The aim of the present retrospective study was to assess the short-term results of pericardial fluid decompression procedures combined with prolonged drainage in patients diagnosed with LPE.

## 3. Material

This study included consecutive patients with LPE who had been treated with pericardial fluid drainage between 2007 and 2017 in the National Tuberculosis and Lung Diseases Research Institute. Despite the retrospective mode of analysis, all the patients with LPE were diagnosed and treated according to the algorithm used in our clinic since 2005. This project was approved by the Institutional Bioethics Committee.

## 4. Methods

### 4.1. Diagnosis of LPE and Cardiac Tamponade

#### 4.1.1. Echocardiography

Echocardiographic examination was used to confirm the presence of LPE and the signs of cardiac tamponade. It was performed by Toshiba Aplio Artida ultrasound device using a sector transducer of 2–5 MHz.

Standard views along with two-dimensional (2D) echocardiographic and Doppler analysis were routinely applied in the assessment of pericardial effusion. Four standard views were used for reproducibility: subcostal view, four-chamber view, and two parasternal views on the long and short axes.

In urgent situations, a single view (usually the subcostal one) was enough to diagnose significant pericardial effusion (Figure 1a).

The volume of PF was assessed according to generally accepted guidelines. The measurements were performed in 2D imaging, in the parasternal long-axis view, at the end of diastole, behind the posterior wall of the left ventricle, and orthogonally both to the myocardial walls and pericardium. For loculated effusions, several off-axis views were used [1,2,3].

Large pericardial effusion was defined as the layer of 20 mm or more of echo-free space behind the posterior wall of the left ventricle.

Threatening cardiac tamponade was recognized when the collapse of the free wall of the right ventricle in the early diastole was observed on echocardiography [1,2,3].

Additional signs of threatening cardiac tamponade included the dilatation of the inferior vena cava (IVC) exceeding 21 mm, the decrease in the physiological collapsibility of the IVC during inspiration to less than 50%, and the dilatation of hepatic veins [1,2,3].

In these patients, the assessment of respiratory variations of the trans-valvular flows by pulsed-wave Doppler echocardiography was performed [1,2,3].

Cardiac tamponade was suggested by reduction in the mitral peak E-wave velocity during inspiration of at least 25% and/or the peak E-wave velocity for tricuspid valve reduction of at least 50% during expiration compared to inspiration [1,2,3].

#### 4.1.2. Radiologic Methods of LPE Diagnostics

Chest X-ray was performed in all the patients: an enlarged cardiac silhouette was shown in all of them (Figure 1b). Other methods of radiologic LPE visualization were applied mostly in patients who did not require immediate pericardial fluid drainage. Chest computed tomography (CT) was performed in 35 patients and chest magnetic resonance imaging (MRI) in 3. 

##### Chest CT

Chest CT examination was performed using a 128-slice GE Revolution GSI scanner (General Medical Equipment SAL, Bauchrieh, Lebanon). The examinations were usually provided monophasically with the administration of iodine contrast agent in an amount adjusted to the patient’s weight. A contrast administration delay of 25 s was used for routine chest examination. In the case of emergency, CT examinations were often performed in a CTPA protocol (CT Pulmonary Angiography) with a smart-prep or bolus test technique. The contrast was not used in the patients with renal insufficiency, iodine sensitization, or a history of allergic reaction to contrast (Figure 2a,b).

##### Chest MRI

Magnetic resonance examinations of the chest were performed by Siemens Avanto 1.5T scanner (Siemens, Munich, Germany). Routine MR sequences and ECG-gated sequences were used and T1-weighted and T2-weighted and fat saturation images were acquired. The examination was performed with gadolinium contrast agent administered through a double syringe system Optistar SF (Optistar, San Francisco, CA, USA) (Figure 3a,b).

### 4.2. Methods of Pericardial Fluid Drainage

#### 4.2.1. Pericardiocentesis and Pericardial Fluid Drainage

Pericardiocentesis was performed with transcutaneous introduction of a needle into the pericardial cavity. It was a bedside procedure monitored with echo imaging, ECG recording, and the patient’s clinical condition.

For pericardial cannulation following pericardiocentesis, the Seldinger technique was used, with a catheter of 2.2 mm 14 Ga and a length of 16 cm. The correct position of both the needle and the catheter was assessed only by echocardiography. The catheter was secured to the skin with 4/0 silk sutures, covered with a sterile dressing [1,2,3,9].

Active manual suction of all pericardial fluid was performed daily, until the fluid return rate was less than 50 mL/day for two consecutive days, and the amount of pericardial fluid behind the left ventricle in standard projections was less than 5 mm, on echocardiography. The pericardial catheter was flushed with isotonic saline after each drainage.

#### 4.2.2. Pericardiotomy/Pericardioscopy and Pericardial Fluid Drainage

Surgical treatment (pericardiotomy, pericardioscopy) was performed by a thoracic-surgeon in the operating room under general anesthesia [1,2,3].

The indications for surgical treatment were:Large pericardial effusion in patients, in whom the etiology of pericardial effusion had not been established yet;Suspicion of malignant pericardial involvement;No safe option to perform a percutaneous pericardial puncture (unfavorable location of fluid, loculations of the fluid, significant obesity, or significant deformation of the chest).

In the last few years, substernal pericardiotomy combined with pericardioscopy has been the preferable option of surgical treatment in our institution.

The surgeon assessed the appearance of pericardial walls and retrieved pericardial fluid and pericardial specimens for histopathological examination. From a separate cut or through a wound, a Pezzer’s drain was inserted into the pericardial cavity, and pericardial drainage was conducted.

The pericardial drain was removed from the pericardial space when the volume of daily drained pericardial fluid was less than 50 mL/day for two consecutive days and the amount of pericardial fluid behind the left ventricle in standard projections was less than 5 mm.

Antibiotic prophylaxis of bacterial infection with ceftriaxone (2 g daily intravenously) was recommended in all of the patients, who underwent pericardiocentesis or pericardiotomy followed by insertion of pericardial catheter, although it was not obligatory.

### 4.3. Pericardial Fluid and Pericardial Specimen Diagnostics

#### 4.3.1. Pericardial Fluid Diagnostics

Pericardial fluid was assessed for:Glucose, protein, LDH, and cholesterol concentration;Morphologic count (leukocytes and erythrocytes);Cytologic examination;Cultures for aerobic and anaerobic bacteria, mycobacteria, and fungi;Polymerase chain reaction (PCR) for *Mycobacterium tuberculosis* infection;Adenosine deaminase (ADA) activity.

#### 4.3.2. Histopathological Analysis of Pericardial Specimens

Pericardial biopsy samples were conducted routinely for histologic examination.

The sections were stained with hematoxylin and eosin, periodic acid Schiff’s with and without diastase, alcian blue, and mucicarmine.

### 4.4. Specific Causes of LPE—Diagnostic Criteria

#### 4.4.1. Neoplastic Pericardial Effusion (NPE)

The diagnosis of NPE was based on positive pericardial fluid cytology and/or the presence of the malignant infiltration in the pericardium biopsy specimen.

#### 4.4.2. Tuberculous Pericardial Effusion

Confident:Positive direct staining of pericardial fluid or pericardial biopsy specimens for mycobacteria and positive PCR test for *M. tuberculosis* of pericardial fluid;Positive result of pericardial fluid or pericardial biopsy culture for *M. tuberculosis*;Caseating granulomas in pericardial biopsy and positive PCR test for *M. tuberculosis*.

Probable:Active tuberculosis of another organ, confirmed with positive culture and lymphocytic pericardial effusion with increased activity of ADA.

#### 4.4.3. Purulent Pericardial Effusion

The purulent character of the fluid was confirmed by physical and biochemical fluid examination (turbid characteristics, low glucose level, high LDH, and high WBC count with an increased number of neutrophils) as well as pericardial fluid culture.

### 4.5. Principles of Intrapericardial Treatment 

#### 4.5.1. Intrapericardial Administration of Cisplatin in a Large Neoplastic Pericardial Effusion

After the confirmation of neoplastic etiology pericardial involvement, intrapericardial cisplatin was given in two schemes:5 × 10 mg cisplatin. Ten milligrams of cisplatin was dissolved in 20 mL of 0.9% NaCl and administered by catheter or drain into the pericardial sac for 5 consecutive days. Drain clamp was removed after 24 h before another dose of cisplatin was applied.1 × 50 mg cisplatin. Fifty milligrams of cisplatin was dissolved in twenty milliliters of 0.9% NaCl and administered once by catheter or drain into the pericardial sac. Drain clamp was removed after 24 h.

#### 4.5.2. Intrapericardial Fibrinolysis in Purulent Pericarditis

##### Streptokinase

Three doses of 500,000 IU of streptokinase were dissolved in 50 mL of 0.9% NaCl and were given every 12 h. After each administration of the drug, the drain was clamped for 12 h.

##### Recombinant Tissue Plasminogen Activator (r-tPA)

In our scheme, 20 mg r-tPA diluted in 100 mL normal saline was given once. The drain was clamped for 24 h.

### 4.6. Criteria of the Short-Term Effectiveness and Safety of Applied Treatment

The short-term effectiveness of therapy was defined as a significant decrease in pericardial fluid volume (determined by echocardiography as less than 5 mm of fluid behind the left ventricular posterior wall) during the hospitalization.

The safety of therapy was assessed by the number of complications such as infection, arrhythmia, respiratory failure requiring respiratory treatment, clinically relevant bleeding, pneumothorax, and myocardial injury.

## 5. Statistics

Statistical analyses were made with using the SPSS Statistica 21.0 statistical package.

The values were presented as medians and ranges. Comparison between two groups for continuous variables was assessed with a Student’s *t*-test and with a Pearson’s χ^2^ for categorical variables. *p* < 0.05 was considered a significant difference.

## 6. Results

The analysis included 74 patients (33 female and 41 male), mean age 58 years (18 ± SD).

In 25 patients, neoplastic pericardial effusion (NPE) was recognized in the course of lung cancer in 15 patients, lymphoma in 2, and breast cancer, thyroid cancer, renal cancer, colon cancer, and osteosarcoma in 1 patient each. In three patients, the pericardial fluid contained neoplastic cells (adenocarcinoma) with unknown primary tumor location.

In 49 patients, non-malignant pericardial effusion (non-NPE) was diagnosed as idiopathic, viral pericarditis in 30, uremic pericarditis in 5, purulent pericarditis in 4, combined with hypothyreosis in 3, collagen tissue disease in 2, other autoimmune cause in 2, and tuberculous pericarditis in 1. In the remaining two patients, the cause of LPE was not known.

In 26 out of 74 patients with LPE, urgent pericardial decompression due to clinical symptoms of cardiac tamponade was performed. This group consisted of 17 with NPE (10 treated with pericardiocentesis and 7 with a surgical method) and 9 patients with non-NPE (3 patients treated with pericardiocentesis and 6 with a surgical method).

In 48 patients (8 with NPE and 40 with non-NPE), LPE without symptoms of cardiac tamponade was diagnosed. The procedures of pericardial fluid drainage used in this group were pericardiocentesis in 5 patients (4 NPE, 1 non-NPE) and surgical treatment in 43 patients (4 NPE and 39 non-NPE).

The details concerning the types of procedures used are included in Figure 4. 

The median drainage duration was 13 days (IQR 6) in the NPE group and 7.5 days (IQR 5) in the non-NPE group (*p* = 0.00001). 

The median volume of drained fluid during decompression was 837 mL (IQR 500) in the NPE group and 775 mL (IQR 637) in the non-NPE group (in purulent pericardial effusion it was 900 mL and 750 mL in tuberculous pericardial effusion). On the following days the median total volume of drained fluid was 650 mL (IQR 878) in the NPE group and 370 mL (IQR 404) in the non-NPE group, *p* = 0.013 (in purulent pericardial effusion, it was 2135 mL and 650 mL in tuberculous pericardial effusion) (Table 1).

The short-term effectiveness of therapy (less than 5 mm of fluid behind the left ventricular posterior wall in echocardiography and no recurrences of large pericardial fluid requiring decompression during the hospitalization) was obtained in all patients.

Non-infective complications were observed in 12 patients (16%), including atrial fibrillation (7 patients), supraventricular arrhythmia other than atrial fibrillation (1 patient), respiratory failure necessitating treatment with mechanical ventilation (2 patients), bleeding complication requiring transfusion (1 patient), pneumothorax requiring drainage (1 patient) (Table 2).

All of these complications concerned patients treated surgically; no such complications were observed in patients who underwent pericardiocentesis.

In one out of four patients with purulent pericardial effusion, earlier opening of the pericardial catheter was needed due to the extensive leak of pericardial fluid (next to the drain).

Infective complications were assessed in 69 patients. Five patients were excluded due to the recognition of bacterial infective pericarditis (purulent in four patients and tuberculous in one) before the drainage was applied. Infective complications of the diagnostic and therapeutic procedures were reported in 7 (10%) out of 69 patients (3 NPE and 4 non-NPE).

### 6.1. Results of Intrapericardial Administration of Cisplatin in NPE

Out of 25 patients with NPE, 17 were treated with intrapericardial instillation of cisplatin (10 patients underwent pericardiocentesis, and cisplatin was given into pericardial space via catheter), 7 patients underwent pericardiotomy, in which cisplatin was applied into pericardial space via Petzzer’s drain. Nine patients received cisplatin with a 1 × 50 mg schedule. However, in one patient, the administration of cisplatin was repeated after 13 days due to persisting pericardial drainage. Eight patients received cisplatin in the 5 × 10 mg schedule. In all patients, a good treatment effect was achieved with no pericardial fluid re-accumulation in the short-term observation time.

### 6.2. Results of Intrapericardial Fibrinolysis in Purulent Pericarditis

Four patients diagnosed with purulent pericardial effusion were treated with pericardial fibrinolysis. Details of the therapy were described previously [10].

### 6.3. Systemic Treatment—Steroids, Colchicine, and NSAIDs

Systemic anti-inflammatory therapy was used in 34 patients (Table 1). Colchicine (0.5 mg × 2 per day if >70 kg or 0.5 mg once daily if ≤70 kg) was implemented in 18 patients (combined with prednisone in 4 patients and combined with NSAIDs in 2 patients) [1,2]. Prednisone (0.5–1 mg/kg per day) was used in 14 patients, including 4 with colchicine [1,2].

Ibuprofen (200–600 mg × 3 per day) was administered to eight patients, including to with colchicine [1,2].

## 7. Discussion

Large pericardial effusion requires invasive treatment in most clinical situations. The choice between pericardiocentesis and surgical procedures, such as pericardiotomy or pericardioscopy, depends on hemodynamic indices, location of pericardial fluid, and the most probable cause of pericardial effusion [1,2,8,9,11].

The recognition of cardiac tamponade, or echocardiographic features of threatening cardiac tamponade, requires immediate pericardial drainage as a lifesaving procedure [1,2,12]. This clinical situation concerned 35% of the patients included in the present study. Pericardiocentesis was performed in 50% of the patients and the surgical procedure in another 50%.

Cardiac tamponade was significantly more frequent in NPE comparing to non-NPE (68% versus 18%); nevertheless, in purulent pericarditis, it concerned two out of four patients (50%). Thus, the indications to immediate heart decompression were most frequent in the NPE as well is in purulent pericardial effusion.

In the group of LPE without the signs of cardiac tamponade, most patients (90%) received surgical treatment. This procedure requires operating room conditions, appropriate equipment, and operator’s skills. The thoracic surgeon performs this procedure in our institution.

The choice of surgical procedure of treatment should be considered especially in those patients in whom the cause of LPE is not known. On such an occasion, the histopathological examination of the pericardial specimen may add information concerning the cause of the pericardial effusion. Currently, the optimal method is substernal pericardiotomy combined with pericardioscopy [13]. In the course of pericardioscopy, a camera is introduced into the pericardial cavity, which allows for the collection of biopsy samples from the pericardial sac under visual control [13].

It is very important to perform chest CT before the surgical procedure in the group of patients who do not require immediate pericardial decompression. CT provides important information, allowing the detection of pericardial loculations, cysts, pericardial thickening and masses, and associated chest abnormalities [1,14]. Thus, it may help to determine the possible cause of LPE, as well as the feasibility of surgical drainage. Moreover, it provides valuable information on the composition of the fluid based on the attenuation values (Hounsfield units (HU)) [1,14]. Attenuation values < 10 HU denote transudative fluids, while those >10 HU indicate exudative ones. Values between 20 and 60 HU suggest a purulent or malignant cause and values >60 HU suggest hemorrhagic fluid. In cases of chylopericardium, values of −60 to −80 HU are reported [1,14]. In our group of patients, CT was performed before the intervention in 35 patients. Chest MRI was rarely performed in our institution in the diagnostics of LPE. The long duration of the study limited its applicability in patients who were unstable or required monitoring. In the present study group, it was performed in only three cases.

The most important question not addressed by recent ESC guidelines nor current literature concerns the role of pericardial fluid drainage in the management of LPE. Our experience indicates that pericardial fluid drainage is of extreme importance in NPE because the re-accumulation of pericardial fluid is a frequent clinical event in patients with NPE who were treated with only pericardiocentesis [5]. Moreover, the prolonged drainage in this group of patients enables the examination of several aliquots of pericardial fluid for the presence of neoplastic cells [1,2,5]. After confirmation of NPE, the direct intra-pericardial chemotherapy is possible via pericardial drain [1,2,15,16,17,18,19,20].

For many years in our center, patients with NPE have been treated with intrapericardial instillation of cisplatin with very good results [18,19,20]. In the present study group, 17 out of 25 patients with NPE (68%) were treated with intra-pericardial instillation of cisplatin in two schemes, depending, among other things, on the clinical condition of the patient, comorbidities, and parameters of renal and hepatic function. In all cases, excellent short-term results were achieved in which there was no fluid re-accumulation and no recurrence of cardiac tamponade.

Pericardial fluid drainage is also obligatory in patients with purulent pericarditis [1]. The treatment of purulent pericarditis remains a challenge for clinicians due to high mortality [1]. In our publication on this rare type of pericarditis, we proposed a scheme of intrapericardial fibrinolysis based on a series of four cases [10].

In endemic areas, 60% of LPE may be caused by tuberculosis, but it is rare in non-endemic areas [21,22]. In our study group, tuberculous pericarditis was recognized in only one patient [23]. Prognosis in patients with tuberculous pericarditis depends on early recognition and initiation of treatment [24,25].

The prolonged pericardial drainage may be complicated by an infection. In our study group, the median duration of drainage was 13 days in NPE and 7.5 day in non-NPE. Infective complications were diagnosed in 10% of patients.

Several limitations of our study should be underlined:This was a retrospective, observational study, making it inherently subjected to confounding bias.No randomization was used to compare various types of pericardial decompression, nor various schemes of intrapericardial cisplatin therapy.The material of the study reflects a single-center population of patients; thus, the specific types of LPE—among others—and post cardio-surgical procedures and complications were not present in the study group.

## 8. Summary

We would like to emphasize the special role of prolonged pericardial drainage in the management of LPE. In our opinion, pericardial drainage seems to be the method of choice for patients in which pericardiocentesis or pericardiotomy is performed, especially when malignant pericardial involvement or purulent pericarditis is suspected. Indwelling pericardial catheters allow for the repetition of cytopathological examination of the fluid and intrapericardial administration of different agents (cisplatin in NPE and rt-PA/streptokinase in purulent pericarditis). It should be pointed out that both the effectiveness and safety of prolonged drainage were very good.

## Figures and Tables

**Figure 1 diagnostics-12-01453-f001:**
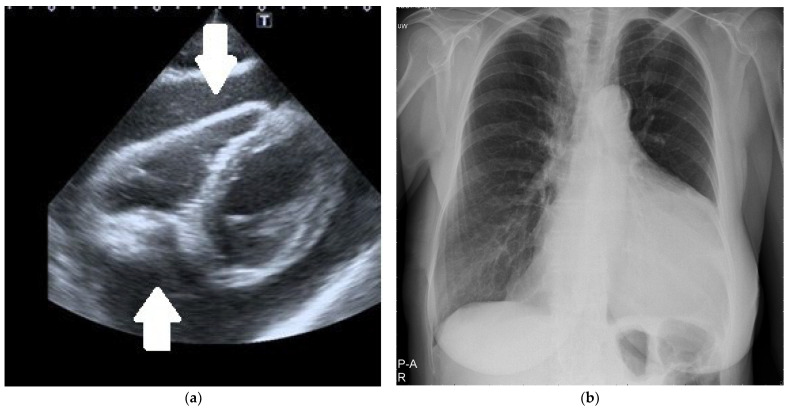
(**a**) Two-dimensional echocardiography. The subcostal intermediate off-axis view shows significant circumferential pericardial effusion (arrows). (**b**) Chest X-ray. Significant heart silhouette enlargement due to a large pericardial effusion.

**Figure 2 diagnostics-12-01453-f002:**
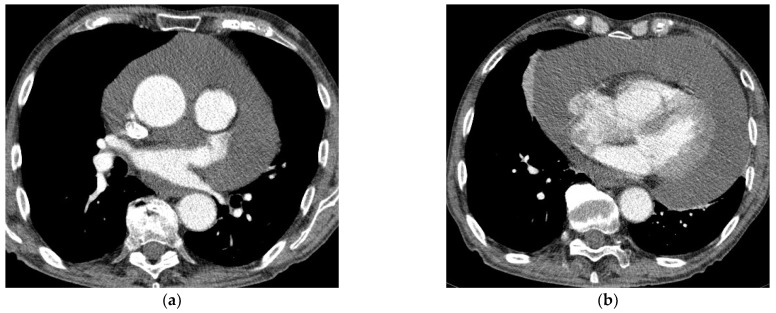
(**a**,**b**) Chest CT with contrast enhancement shows a large pericardial effusion.

**Figure 3 diagnostics-12-01453-f003:**
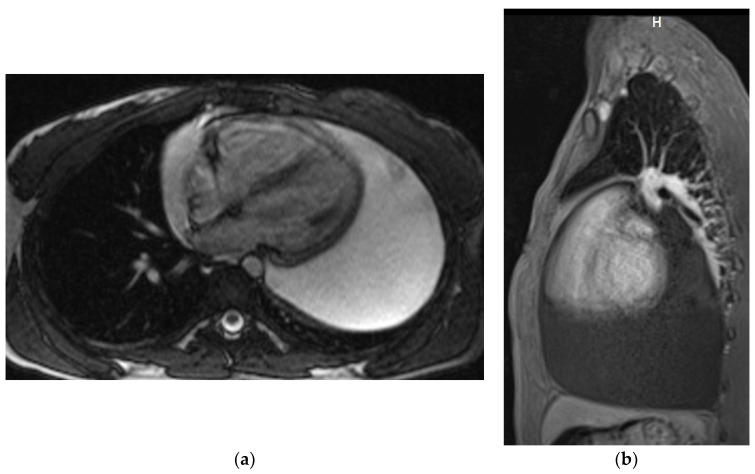
Chest MRI. T2-weighted image, axial view (**a**) T1 weighted image with fat saturation and with contrast enhancement, sagittal view (**b**). Large pericardial effusion is seen, hyperintense on T2-weighted image (**a**) and hypointense on a T1-weighted image (**b**).

**Figure 4 diagnostics-12-01453-f004:**
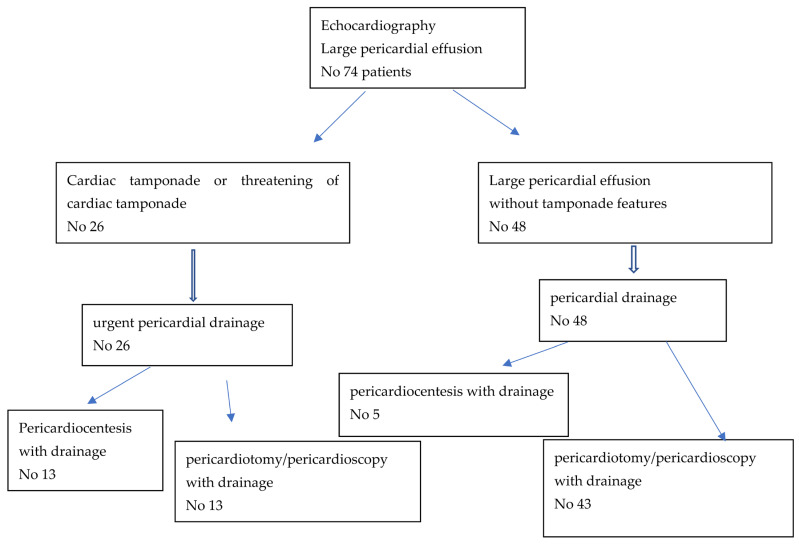
Types of procedures used in patients with large pericardial effusion.

**Table 1 diagnostics-12-01453-t001:** Etiology, hemodynamic consequences, and applied treatment modalities in 74 with large pericardial effusion.

Type of Pericarditis No	Neo No 25	Non-NPE		
		Other No 44	Purulent No 4	TBC No 1
Cardiac tamponade No (%)	17 (68 %)	7 (16%)	2 (50%)	0
	*p* = 0.001			
Pericardiocentesis No (%)	14 (56%)	4 (9%)	0	0
	*p* = 0.0001			
Surgical treatment No (%)	11 (44%)	40 (90%)	4 (100%)	1
	*p* = 0.02			
Intrapericardial treatment No (%)	Cisplatin 17 (68%)	0	r-tPA4 (100%)	0
Total amount of drained fluid during the intervention mL	837 mL	775 mL	900 mL	750 mL
Median volume after intervention mL	650 mL	370 mL	2135 mL	650 mL
	*p* = 0.013			
Systemic treatment with colchicine No (%)	4 (16%)	14 (32%)		
Systemic treatment with prednisone No (%)	4 (16%)	10 (23%)		
Systemic treatment with NSAIDs No (%)	0	8 (18%)		

**Table 2 diagnostics-12-01453-t002:** Rate of complications in the group of patients treated with surgical intervention compared to pericardiocentesis.

	Pericardiotomy/Pericardioscopy No 56	Pericardiocentesis No 18
Paroxysmal AF	7 (12%)	0
SVT	1 (2%)	0
Respiratory failure requiring ventilation support	2 (3%)	0
Clinically relevant bleeding	1 (2%)	0
Pneumothorax	1 (2%)	0
Infective complications	5 (9%)	2 (11%)
Total	17 (30 %)	2 (11%)

## Data Availability

Data supporting reported results can be found in source data collected in National Tuberculosis and Lung Diseases Research Institute.

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
