# Peer review of "Large Pericardial Effusion—Diagnostic and Therapeutic Options, with a Special Attention to the Role of Prolonged Pericardial Fluid Drainage"

_diagnostics, 2022, doi:10.3390/diagnostics12061453_

Round 1
Reviewer 1 Report
There is a relevant clinical research, slightly limited by retrospective model of the study.
Concerning "Methods" :
The description of the therapy details of steroids (prednison), of colchicine, of ceftriaxone (given as prophylaxis) and of NSAIDs is missing. I mean the doses and the time of the therapy e.g.
Concerning language:
I propose the authors to correct the punctuation, especially the spaces and the articles as well as. I propose do not start the sentence with the numeral.
My proposals for some style changes are:
66: ...pericarditis and refers to 60% of LPE..
91: .., used in our clinic since 2005.
105: In urgent situations,...
144: ..study performence ....
315,316: In one case of 4 patients with purulent pericardial effusion, due to the extensive leak of pericardial fluid (next to the drain from the catheter), earlier opening was needed, after ..
357, 358: This procedure requires operating room conditions, appropriate equipment and operator's skills. The thoracic-surgeon performs this procedure in our institution.
365 and 366: It is very important to perform chest CT before surgical procedure in the group of patients who do not require immediate pericardial decompression.
372: .....suggest a purulent or malignant cause and..
396: The prolonged pericardial drainage may be complicated by the infection.
Thank you
Author Response
Dear Reviewer,
thank you very much for your comments. We made corrections as suggested. We hope you enjoy the article more readable now.
Yours faithfully,
The Authors

Reviewer 2 Report
There are gross spelling and grammatical errors just in the abstract alone.
As an example:
comma after effusion on line 35
line 47, either use women and men or, better, switch to male / female
line 48 "patients presented with cardiac tamponade symptoms"
lines 54 and 55 need to be reworded completely
These are very obvious errors. In certain situations, sentences are poorly worded and do not make sense. I would recommend proofreading the article again, correcting all of the English spelling and grammatical errors, and resubmitting.
Author Response

(The authors gave the same response as above.)

Reviewer 3 Report
-
Appreciate authors effort in writing this manuscript. They described single center experience of large pericardial effusion management, extended drainage safety and efficacy.
43 out of 48 patients with LPE/no tamponade group underwent pericardiotomy/pericardioscopy, while only 5 patients in LPE/no tamponade group underwent pericardiocentesis. Pericardiocentesis is the most commonly used management approach for LPE. Authors did good job in describing various reasons why pericardiotomy/pericardioscopy was performed in these patients.
-
Few minor mistakes:
-
49 In 17 out of 25 patients with neoplastic pericardial effusion intrapericardial instillation of
-
50 cisplatin was inicated.
-
54 Pericardial fluid drainage for patients required pericardial decompression due to LPE was
-
55 saftey and efficacy and enables intrapericardial treatment.
-
126 Chest X-ray was performed in all of the patients. In most cases in showed enlarged
-
166 following pericardiocentesis, the Seldinger technique was used, with the catheter of 2,2
Author Response

(The authors gave the same response as above.)

Reviewer 4 Report
The paper is interesting. It requires some clarifications.
Minor spelling and check of the English language is required: See for examples:
- in the text all "figure" and "table" should be "Figure" and "Table"
- line 48 "symptopms"
- line 50 "inicated"
- line 55 "saftey" "efficacy" : perhaps it shoud be better "safe and effective"
- line 103 instead of "projections" it could be better "views"
- In Table 1: instead of "prednison" should be "prednisone".
- In Table 2: instead of "Sum" probably better "Total".
- line 323: "of of"
- line 396: "complicated infection", probably better: "complicated by infection".
- line 408: "extended": probably better "prolonged".
Line 100: the types of used transducers should be indicated.
Figures seems too many. Probably a composite Figure could include Figure 1 and 2 as Figure 1A and 1B. The same for Figure 3 and 4 as composite Figure 2A and 2B.
For the legend of Figure 1 (lines 108-109) I would suggest to change: "2D Echocardiography. The subcostal intermediate off-axis view shows significant circumferential pericardial effusion (arrows)". Arrows should be added.
Line110-112: it should be indicated (with references) where and how linear and volume measurements were performed. For linear measurements it should be also indicated if they were performed orthogonally both to the myocardial walls and to the pericardium.
Lines 115-116: it is not clear if the measure of paradoxical pulse was also assessed clinically ad included in the diagnosis of tamponade. References for the diagnosis of cardiac tamponade should be also reported.
Line 126: "in most cases": how many cases ? How many cases performed also CT and MRI and at what time during the clinical course ?
Lines 162-169: the method of drainage of the large amount of fluid should be described in more detail: all fluid drained immediately by active suction ? fall drainage? Contrast fluid was injected into the pericardial sac to check needle position ? References should be added.
Line 209: pericardial specimens were only obtained by surgical approach or also by needle in some cases?
Lines 231-233: References should be added.
Lines 240-246: References should be added.
Lines 251-255: References should be added.
Lines 338-341: Dosages, duration of treatments and References should be added.
Author Response

(The authors gave the same response as above.)

Round 2
Reviewer 2 Report
Significant English errors remain. Here are just a few of them:
include STD for mean values and IQR for median
lines 57-58 need rewording
lines 63 and 64 need to be reworded
lines 83-85 need to be reworded
line 99 needs to be reworded "proceeded" is incorrect here
lines 121-126, run on
line 141, enlarged not enlarge
reword 152-153
155 - slice not slices
264 in a large instead of in large
268 for instead of on
279 were instead of where
There are others as well in the second half of the manuscript.
Author Response
Dear Reviewer,
thank you very much for your comments. We made corrections as suggested. We hope that the article will become more readable now.
Yours faithfully,
The Authors
Comments and Suggestions for Authors
Significant English errors remain. Here are just a few of them:
include STD for mean values and IQR for median Done, I am very thankful for the suggestion.
lines 57-58 need rewording Done, I am very thankful for the suggestion.
lines 63 and 64 need to be reworded Done, I am very thankful for the suggestion.
lines 83-85 need to be reworded Done, I am very thankful for the suggestion.
line 99 needs to be reworded "proceeded" is incorrect here Done, I am very thankful for the suggestion.
lines 121-126, run on Done, I am very thankful for the suggestion.
line 141, enlarged not enlarge Done, I am very thankful for the suggestion.
reword 152-153 Done, I am very thankful for the suggestion.
155 - slice not slices Done, I am very thankful for the suggestion.
264 in a large instead of in large Done, I am very thankful for the suggestion.
268 for instead of on Done, I am very thankful for the suggestion.
279 were instead of where Done, I am very thankful for the suggestion.
There are others as well in the second half of the manuscript.
Reviewer 4 Report
Lines 63-64 should be changed: "Pericardial fluid drainage for patients requiring pericardial decompression due to LPE was a safe and effective method, which also enables further intrapericardial treatment."
Line 109: "using a sector transducer" of how many MHz ?
In Figure 1a arrows should be added.
Line 141-142: " Chest X-ray was performed in all the patients: an enlarged cardiac silhouette was shown in all of them.".
Line 152: " CT was performed in 35 patients..."
Line 333: ....." and applied treatment modalities..."
Line 339: "Observed non-infective complications: ...
Author Response
Dear Reviewer,
thank you very much for your comments. We made corrections as suggested. We hope that the article will become more readable now.
Yours faithfully,
The Authors
Comments and Suggestions for Authors
Lines 63-64 should be changed: "Pericardial fluid drainage for patients requiring pericardial decompression due to LPE was a safe and effective method, which also enables further intrapericardial treatment." Done, I am very thankful for the suggestion.
Line 109: "using a sector transducer" of how many MHz ? Done, I am very thankful for the suggestion.
In Figure 1a arrows should be added. Done, I am very thankful for the suggestion.
Line 141-142: " Chest X-ray was performed in all the patients: an enlarged cardiac silhouette was shown in all of them." Done, I am very thankful for the suggestion.
Line 152: " CT was performed in 35 patients..." Done, I am very thankful for the suggestion.
Line 333: ....." and applied treatment modalities..." Done, I am very thankful for the suggestion.
Line 339: "Observed non-infective complications: ... Done, I am very thankful for the suggestion.